# Exploring the Coupling Coordination and Key Factors between Urban–Rural Integrated Development and Land-Use Efficiency in the Yellow River Basin

**Caiting Shen [1], Linna Shi [2], Xinyan Wu [2,3] , Jinmei Ding [2] and Qi Wen [2,4,*]**

[1]   School of Economics and Management, Ningxia University, Yinchuan 750021, China
[2]   School of Geography and Planning, Ningxia University, Yinchuan 750021, China
[3]   College of Geographical Sciences, Qinghai Normal University, Xining 810016, China
[4]   School of Architecture, Ningxia University, Yinchuan 750021, China
[*]   Correspondence: wenq@nxu.edu.cn

**Abstract:** Exploring the complex dynamic relationship between urban–rural integrated development and land-use efficiency can contribute to most efficient urban–rural land-use and the rational promotion of urban–rural integrated development. This study established an evaluation model of urban–rural integrated development, adopted the super-efficiency SBM model to measure land-use efficiency, and studied the evolution of the spatial–temporal patterns of urban–rural integrated development and land-use efficiency coupling in the Yellow River Basin. We also examined the factors affecting them with the help of the coupling coordination degree model, non-parametric kernel density estimation, and geographic probes. The results indicate the following: (1) Within the study period, the coupled coordination of urban–rural integrated development and land-use efficiency was similar to the spatial distribution characteristics of land-use efficiency, both showing a "high at both ends and low in the middle" trend. (2) The coupled coordination increased over time; however, a lagging land-use efficiency was a crucial impediment to improving the coupling coordination degree. (3) Carbon emissions, urbanization rate, and per capita GDP were key drivers. The results of this study can provide a reference for local governments in the Yellow River Basin and other similar areas to propose paths to optimize the allocation of urban and rural land-use.

**Keywords:** urban–rural relationship; urban–rural integrated development; land-use efficiency; coupling coordination relationship; geographic detector

## 1. Introduction

In the rapid global industrialization and urbanization process, urban–rural polarization is evident in many countries around the world. It is accompanied by problems such as "urban diseases" and "hollowing out of the countryside" [1,2]. The orderly integration and balanced development of cities and villages is not only a cornerstone of social stability but it is also closely related to the realization of SDG 10 (reduce inequality within and among countries) and SDG 11 (make cities and human settlements inclusive, safe, resilient, and sustainable) [3,4]. For a long time, China's dual system with an urban–rural division has resulted in an imbalance between urban and rural development and the allocation of land elements, and this imbalance has become a key issue in China's new era of high-quality development [5,6]. To facilitate the bi-directional mobility of resources between urban and rural regions in China, the Chinese government has proposed establishing a robust institutional framework and policy structure for urban–rural integration [7]. The land is the physical carrier of the two settlement spaces, urban and rural [8]. However, under the current non-market mechanism of China's land transaction model, many land-use issues such as severe wastage of land resources and low land-use efficiency (LUE), have already seriously constrained urban–rural integrated (URI) development [9,10]. To protect China's

existing land resources and alleviate the contradiction between land supply and demand, China has proposed the conservation and intensive use of land resources. At the same time, it has further pointed out that promoting a fundamental change in how land resources are used is necessary [11]. Along with introducing China's pilot comprehensive reform policy on the market-based allocation of land transactions, URI development and LUE have become more closely linked. On the one hand, urban–rural integration improves the efficiency of land resource allocation through the smooth circulation of land resources between urban and rural areas, which, in turn, promotes efficient land-use; on the other hand, the economic and intensive use of land improves the comprehensive efficiency of land-use, which can effectively alleviate the contradiction between population growth and exceedance of the carrying capacity of the land, accelerate urban–rural population mobility, and effectively promote URI development [12,13].

The urban–rural relationship is the most fundamental economic and social relationship, and most countries in the world have been exploring the structure of the urban–urban relationship in urbanization as appropriate for their national conditions [2,14,15]. Research on urban–rural relationships has mainly resulted in the urban–rural dual structure theory represented by the Ranis–Fei model, the urban–rural coordination theory represented by the core–periphery model, and the urban–rural integration theory described by the Marxist theory of urban–rural relationships [16–18]. China is currently in a critical period of transforming the urban–rural ties, and the "urban–rural integrated development" plan proposed by the Chinese government in 2017 is an important initiative to solve a series of problems such as the division of urban and rural areas, the division of land, and the separation of people and land, and to create a new type of urban–rural relationship [19,20]. Currently, the research on urban–rural integration focuses on the theoretical connotation, the construction of an indicator system, the change in spatial and temporal patterns, and the influencing factors [21,22]. Jiang [21] constructed a multi-level urban–rural integration evaluation index system at the population, land, and economic levels. Based on the quality of life perspective, Ma [17] created an assessment system for urban–rural integration from economic, social, and environmental perspectives. Silva [23], using an integrated research methodology approach based on ecological and socio-economic factors in the Paraíba Valley (Brazil), found that rural–urban coupling enhances synergies between rural and urban areas and can promote the sustainability of arable land and improve ecological services. The study found that long-standing urban–rural development imbalances have widened the gap [16–18,24]. Sánchez-Zamora [25] studied the region of Andalusia (Spain) and found that the financial crisis has severely exacerbated regional and rural–urban inequalities, but that employment and entrepreneurship, economic diversification, and technological upgrading have helped to raise the level of rural development, thereby reducing the rural–urban gap. Furthermore, precise poverty alleviation, green growth, and the digital economy positively affected urban–rural integration [26–28].

Improving LUE can effectively promote integrated urban–rural development and is significant for achieving the global Sustainable Development Goals (SDGs) [3,29,30]. Much research has focused on this, mainly resulting in theories of the urban spatial structure, represented by concentric circles, sectors, and multiple nuclei, with the intelligent growth theory emphasizing the efficient use of existing land, and the theory of compact cities [31,32]. The research has mainly focused on defining the connotations of LUE, constructing an indicator system, analyzing the spatial and temporal patterns, and exploring the paths for improvement [33,34]. Some scholars have used the DEA model, super-efficiency SBM model, panel data regression model, and various hybrid models to measure LUE levels. Wang [35] constructed an LUE evaluation index system by taking industrial "three waste" emissions as "non-desired outputs" and found that urban LUE has different impacts on the optimization of industrial structure in various provinces and cities in China. Haller [36] studied urban-rural land change in the Central Peruvian Andes and found that urban expansion led to a reduction in arable land, which, in turn, lowered the incomes of farmers. Masini [37] analyzed the relationship between economic growth and LUE in 417 cities

in 17 countries in Europe, and found that the higher the level of the economy, the more efficient the land-use. Song [3] studied LUE by constructing a ratio of land-consumption rate to population growth rates and found a coherent relationship between LUE and the Sustainable Development Goals (SDGs). By building a Tobit regression model, Yu [38] found that the economic level, economic structure, and government regulation positively impacted LUE.

At present, the research on urban–rural relationships and land-use focuses more on unilateral research URI development or LUE. The analysis of the relationship between the two focuses on URI development and land-use transformation, urban–rural spatial evolution and land-use changes, urbanization development and land-use transformation, and rural revitalization and arable land utilization [1,13,39,40]. There are fewer studies on the relationship between URI development and LUE. Niu [12] found that optimizing land-use, including improving LUE level, can restructure the urban–rural socio-economic pattern and promote integrated urban–rural development. By analyzing land-use in Spain, Serra [41] found that the rationalization of land-use can improve land-use efficiency, and thus reconstruct urban–rural relations. Taking Israel as the object of his study, Bittner [42] combined the intensive use of land with the spatial evolution of urban and rural spaces. He found that specialized and intensive land-use improves the efficiency of the land, and ultimately, the urban space interacts with the rural space in a new way. Yin [43] also discovered that LUE can be effectively enhanced through land consolidation and land-use transformation, promoting urbanization, rural revitalization, sustainable regional development, and integrated urban–rural development. Chen [13] used kernel density estimation, spatial autocorrelation analysis, and fixed-effects to study 372 samples from 31 province-level administrative regions in China. The study revealed that, under ideal conditions, land-use transformation can be achieved by enhancing the value of land elements and LUE, ultimately promoting integrated urban–rural development. Wu [44] found that land financing can effectively enhance integrated urban–rural development and thus improve the LUE level. Song [45] conducted a study using panel data from 30 province-level administrative regions in China, spanning the period from 2010 to 2019. The findings indicated that the overall degree of coupled coordination between URI development and LUE was not high, but it increased year by year. The existing studies have provided theoretical and empirical support for the association, interaction, and enhancement path between urban–rural relationships and land-use. However, the existing studies on URI development and LUE have provided few empirical studies on the dynamic relationship between the two. Moreover, there are fewer analyses on the factors affecting them.

The Yellow River Basin (YRB) is a substantial food production base and a significant supply base for energy resources in China. In 2019, the Chinese government pointed out the essential position of the YRB in China's economic and social development and ecological urban–rural integration. However, the YRB faces many problems, such as tightening constraints on land resources, the prominent imbalance between urban and rural development, and poor-quality of economic growth. In-depth exploration of the dynamic relationship between URI development and LUE in the YRB that can reveal the evolution of the spatial and temporal pattern of the coupled and coordinated development of these two aspects and the factors affecting them can provide a reference for optimizing the allocation of urban and rural land-use; at the same time, this study has significant reference value for promoting the cyclic flow of urban and rural resource elements, improving the efficiency of urban construction and arable land resources, and advancing the integrated development of urban and rural areas.

In this context, the study established an evaluation system of URI development indicators in the five dimensions of people, land, economy, society, and ecology. It was used to calculate the urban–rural integration level of 61 prefectures in the YRB using the linear weighting method. At the same time, the super-efficiency SBM model was used to measure the use efficiency of urban construction land and rural arable land, and the integrated LUE was obtained through weighting. In addition, the coupled coordination

degree (CCD) model and non-parametric kernel density estimation method were used to explore the coordinated relationship and dynamic evolution of URI development and LUE in the YRB. Finally, the influencing factors of the coordinated development level of URI and LUE in the YRB were measured with the help of a geographic detector. This article aims to provide empirical and policy references for improving URI development and land resource utilization efficiency in regions similar to the YRB.

## 2. Study Area and Indicator System

### 2.1. Study Area and Data Sources

As one of China's most important economic growth areas, food production bases, and ecological barriers, the YRB faces multiple challenges including limited land resources, unbalanced regional development, and vulnerable environments. The Yellow River flows through 71 cities (including states and leagues) in 9 provinces, and the overall topography is characterized as high in the west and low in the east [46]. For this study, the YRB includes 61 geospatial units due to the missing data of some prefecture-level cities (Figure 1). The YRB was divided into three regions following the principle of "taking the natural Yellow River Basin as the basis and maintaining the integrity of the administrative units at the regional level as far as possible": the upstream region (including Qinghai, Gansu, and Ningxia, with a total of 14 cities), the middle reaches of the Basin (including Shanxi, Shaanxi, and Inner Mongolia, with a total of 26 cities), and the lower reaches of the Basin (including Henan and Shandong, with a total of 21 cities) [11,47,48].

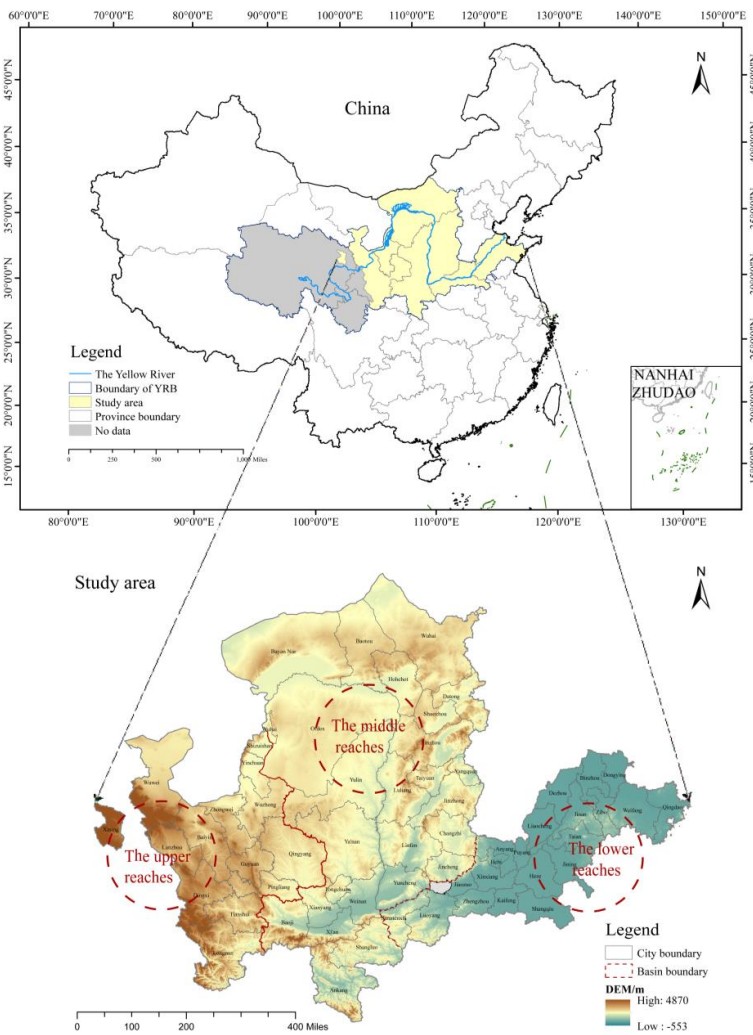

**Figure 1.** The spatial scope of the Yellow River Basin (YRB) in China.

In this study, there were two sources of data used in the paper (Table 1). The statistical data (including economic data, social data, land data, and ecological data) were from the official statistical website, and some of the missing data were filled in by interpolation. The raster data (climatic environmental data) were from the Institute of Resource and Environmental Science and the Data Center of the Chinese Academy of Sciences, and the raster data were all processed by ArcGIS. For all variables expressed in monetary terms, we deflated them using the Consumer Price Index (CPI) for each city with a base period of 1978. This study focused on the period from 2003 to 2021. Since Laiwu in Shandong Province was merged into Jinan in January 2019, and considering the consistency for higher data quality, this study also merged Laiwu into Jinan for calculations.

**Table 1.** Description of data types and sources.

| Type | Date Presentation |
|---|---|
| Economic data | China Urban and Rural Construction Statistical Yearbook (2003–2021); Provincial Statistical Yearbook (2003–2021); EPSDATE (https://www.epsnet.com.cn), accessed on 30 April 2023. |
| Social data | Provincial Statistical Yearbook (2003–2021); EPSDATE (https://www.epsnet.com.cn), accessed on 30 April 2023. |
| Land data | China Urban Construction Statistical Yearbook (2003–2021); EPSDATE (https://www.epsnet.com.cn), accessed on 30 April 2023. |
| Ecological data | Carbon Emissions Accounts and Datasets, CEADs (https://www.ceads.net.cn); Multi-resolution Emission Inventory for China, MEIC (http://meicmodel.org.cn/); EPSDATE (https://www.epsnet.com.cn); accessed on 30 April 2023. |
| Climatic environmental data | Institute of Resource and Environmental Science and the Data Center of the Chinese Academy of Sciences (https://www.resdc.cn/), the resolution of elevation is 30 m, and the resolution of Precipitation is 1 km, accessed on 30 April 2023. |

### 2.2. Design of the Evaluation Indicator System

The URI development is a process, a state, and a goal, determined by a combination of population, spatial, economic, social, and ecological factors. Thus, this study constructed a multidimensional evaluation index system for URI development from these five dimensions [16,27]. The LUE is determined by a combination of natural, economic, and social factors, and this study used the per capita input–output efficiency as the LUE, and the input and output indicators were selected with full consideration of the land's economic, social, and environmental benefits [49,50]. To reflect the overall land resource utilization efficiency, this study selected the input–output indicators of utilization efficiency of urban construction land and rural arable land as the input–output indicators, respectively [20,50].

Combined with the research on the structure of an indicator system for URI development and LUE in existing studies and considering the availability and reliability of the data, indicator systems for evaluating URI development (Table 2) and the LUE (Table 3) were developed.

**Table 2.** Indicator system for urban–rural integrated (URI) development.

| Index Dimensions | Index & Properties | Basic Index | Calculation or Description of the Index & Unit | Interpretation of the Index |
|---|---|---|---|---|
| | X1 (+) | Population mobility rate | Urban population/total population (%) | Population mobility can positively impact the development of the rural economy, creating a beneficial urban–rural flow of people. |

Table 2. *Cont.*

| Index Dimensions | Index & Properties | Basic Index | Calculation or Description of the Index & Unit | Interpretation of the Index |
|---|---|---|---|---|
| People | X2 (−) | Coefficient of contrast between urban and rural employment | Employment of urban households/employment of rural households | Reducing the gap between the incomes and consumption of urban–rural residents, particularly in food, culture, education, recreation, and daily electricity bills, will promote balanced incomes and consumption between urban–rural households. |
| | X3 (−) | The ratio of per capita annual disposable income of urban to rural residents | Per capita annual disposable income of urban households/per capita annual net income of rural households | |
| | X4 (−) | The ratio of per capita income of urban to rural residents | Per capita consumption of urban households/per capita consumption of rural households | |
| | X5 (−) | Comparison coefficient of culture, education, and entertainment between urban and rural areas | Urban residents' household expenditure on culture, education, and entertainment/rural residents' household expenditure on culture, education, and entertainment | |
| | X6 (+) | The ratio of Engel's coefficients of urban to rural households | Engel's coefficient of urban households/Engel's coefficient of rural households | |
| | X7 (+) | The ratio of electricity consumption of urban to rural residents | Urban domestic electricity consumption/rural domestic consumer electricity consumption | |
| Land | X8 (+) | The ratio of urban to rural residential space | Urban residential space/rural residential space | Reflect the allocation and utilization of land resources between urban and rural areas |
| | X9 (+) | Urban spatial expansion | Built-up area/cropland area | |
| | X10 (+) | Land urbanization level | Built-up area/land area (%) | |
| | X11 (+) | Passenger turnover | Total passenger transportation (ten thousand people) | Reflect urban–rural accessibility, the greater the accessibility, the better the integration of urban–rural land. |
| | X12 (+) | Per capita postal and telecommunications services | Total postal and telecommunications services/total population (CNY/person) | |
| | X13 (+) | Regional economic operation condition | GDP per capita (CNY/person) | Under normal circumstances, regions with higher levels of economic development are more able to promote industry to feedback to agriculture and promote urban–rural integration development. |

<div align="center">**Table 2.** *Cont.*</div>

| Index Dimensions | Index & Properties | Basic Index | Calculation or Description of the Index & Unit | Interpretation of the Index |
|---|---|---|---|---|
| Economy | X14 (+) | Agriculture finance | Public expenditure on agriculture, forestry and water resource projects/financial expenditure (%) | Reflects the central and local financial input to rural areas, the greater the input, the more conducive the area is to the URI development. |
| | X15 (−) | Ratio of fixed asset investment in urban–rural areas | Rural fixed asset investment/urban fixed asset investment | Reflects the strength of investments in fixed assets in urban–rural regions, especially in infrastructure improvement and optimization of livelihood projects. |
| | X16 (+) | Binary comparison coefficient | (Output value of primary industry/employees in the primary industry)/(Output value of secondary and tertiary industries/employees in secondary and tertiary industries) | Reflects the difference in economic structure between the traditional agricultural sector and the modern industrial and service sectors; the smaller the industrial gap between urban and rural areas, the more conducive the areas are to promoting URI. |
| | X17 (+) | Agricultural mechanization level | Total power of agricultural machinery/arable land area (Kilowatt/hectares) | Agricultural modernization has a positive impact on rural economic development and URI. |
| Society | X18 (+) | Internet penetration rate | Internet access in urban–rural areas/total number of urban–rural households (%) | Reflect urban–rural residents' access to public services. |
| | X19 (−) | The ratio of the level of medical protection for urban to rural residents | Hospital beds per 1000 population in urban healthcare institutions/hospital beds per 1000 population in rural healthcare institutions | |
| Ecology | X20 (+) | Harmless treatment rate of domestic waste | % | Reflect the level of the living environment for urban–rural residents, harmless treatment of domestic rubbish and sewage treatment can improve the living conditions of residents, and optimize the urban–rural ecological environment which can improve the URI. |
| | X21 (+) | Wastewater treatment | % | |
| | X22 (+) | Industrial sulfur dioxide emissions | Metric tons | Industrial pollution mainly affects the urban environment. |
| | X23 (+) | Industrial wastewater discharge | Metric tons | |
| | X24 (+) | Industrial solid waste emissions | Metric tons | |
| | X25 (+) | Ratio of investment in environmental pollution treatment | Investment in environmental pollution control/total output value (%) | Investment in pollution control represents the level of environmental pollution control, and a high level of control benefits URI. |

**Table 3.** Input–output variables for land-use efficiency (LUE).

| Goal Layer | Criterion Layer | Urban Indicators | Rural Indicators |
|---|---|---|---|
| Inputs | Land | Urban built-up area | Arable land area |
| | Labor force | Construction employees per unit area of building | Labor force per unit area of cultivated land |
| | | Urban residential space | Rural residential space |
| | Energy | Capital investment per unit area of building | Agricultural machinery per unit area of cultivated land |
| Outputs | Expected outputs | Social benefit | Per capita annual disposable income of urban households | Per capita annual net income of rural households |
| | | Economic benefit | The gross output value of the construction industry per unit area of building | Agricultural output per unit of cultivated area |
| | Non-expected Outputs | Emission reduction | Emissions of the "three wastes" (wastewater, waste gas, and industrial solid waste) | — |

## 3. Methods

### 3.1. The Linear Weighting Method for Measuring the Level of Integrated Urban–Rural Development

The entropy weight method is one of the methods in the objective assignment method, which can decide the weight of indicators through the size of the information utility value of the indicators. In the study, the range method was used to process the positive index and negative index due to the differences in the dimensions and magnitudes of the indicators, respectively [1,16].

$$
\begin{cases}
z_{ij}^{+} = \frac{x_{ij} - \min(x_{ij})}{\max(x_{ij}) - \min(x_{ij})} + 0.0001 \\
z_{ij}^{-} = \frac{\max(x_{ij}) - x_{ij}}{\max(x_{ij}) - \min(x_{ij})} + 0.0001
\end{cases}
\tag{1}
$$

In Equation (1), $x_{ij}$ refers to the initial matrix, $z_{ij}^{+}$, $z_{ij}^{-}$ represent the normalized matrices for positive and negative indicators, respectively, and $\max(x_{ij})$ and $\min(x_{ij})$ reflect the maximum and minimum values of initial data, respectively.

The indicator $j$ proportion is calculated as shown as Equation (2), and calculation the information entropy $e_j$ by using the Equation (3); then, the weights $w_{ij}$ for indicator $j$ is calculated with Equation (4):

$$
p_{ij} = \frac{z_{ij}}{\sum_{i=1}^{n} z_{ij}}
\tag{2}
$$

$$
e_j = -\frac{1}{\ln n} \sum_{i=1}^{n} p_{ij} \ln p_{ij}
\tag{3}
$$

$$
w_{ij} = \frac{1 - e_j}{\sum_{j=1}^{m} 1 - e_j}
\tag{4}
$$

where $p_{ij}$ refers to the data proportion, $e_j$ refers to the information entropy, $n$ is the number of the index $i$, $m$ is the number of the indicator $j$, and $w_{ij}$ is the weight matrix derived from the entropy weight method.

Finally, the study used the comprehensive score $U$ to measure the URI level; the comprehensive score $U$ is measured by the linear weighting method, as shown in Equation (5):

$$
U = w_{ij} \times z_{ij}
\tag{5}
$$

### 3.2. The Super-Efficiency SBM Model for Measuring Land-Use Efficiency (LUE)

The LUE is the extent to which the value of inputs such as resources, labour, and capital is realized on the land. The paper measured the efficiency of land-use, using the super-efficiency SBM model containing the non-expected outputs is as follows [29,51]:

$$\rho = \min \frac{1 - \frac{1}{N}\sum\limits_{n=1}^{N} S_n^x / x_{k'n}'}{1 + \frac{1}{M+I}\left(\sum\limits_{m=1}^{M} S_m^y / y_{k'm}^t + \sum\limits_{i=1}^{I} S_i^b / b_{k'i}^t\right)}$$

$$\left\{\begin{array}{l} \sum\limits_{t=1}^{T}\sum\limits_{k=1}^{K} z_k^t x_{kn}^t + S_m^y = x_{k'n}^t, n = 1, \cdots, N \\ \sum\limits_{t=1}^{T}\sum\limits_{k=1}^{K} z_k^t y_{km}^t - S_m^y = y_{k'm}^t, m = 1, \cdots, M \\ \sum\limits_{t=1}^{T}\sum\limits_{k=1}^{K} z_k^t b_{ki}^t + S_i^b = b_{k'i}^t, i = 1, \cdots, N \\ z_k^t \geq 0, S_n^x \geq 0, S_m^y \geq 0, S_i^b \geq 0, k = 1, \cdots, K \end{array}\right\} \tag{6}$$

where $\rho$ is the evaluation value of LUE; $N, M, I$ refers to the number of corresponding input, expected factors, and non-expected output factors respectively; $n, m, i$ are the corresponding indicator types; $x, y, b$ are the types of slack variables; and $S_n^x, S_m^y, S_i^b$ represent the slack vectors of the corresponding input, expected factors, and non-expected factors, respectively.

### 3.3. The Coupled Coordination Degree (CCD) Model for Evaluating the Coupling Coordination Level of Urban–Rural Integrated (URI) Development and LUE

The CCD model is widely used to study the interaction between multiple systems. In the study, the CCD model was used to evaluate the coupling coordination levels of the URI development and LUE, and the calculation formula is as follows [49,50]:

$$\left\{\begin{array}{l} C = \sqrt{(U \times \rho) \big| ((U + \rho)|2)^2} \\ T = \alpha \times U + \beta \times \rho \\ D = \sqrt{C \times T} \end{array}\right. \tag{7}$$

where $C$ refer to the coupling levels, while $U_1$ and $U_2$ represent the level of URI development and LUE, respectively. $T$ is the comprehensive evaluation value, and $D$ is the coupling coordination levels. Generally, the subsystems are considered to be of equal importance and, therefore, $\alpha = \beta = 0.5$. Considering the current circumstances and other experts' research, the coupling coordination degree has been classified into six stages in this study (Table 4). Furthermore, the synchronous development model was implemented to separate the synchronous relationship between URI development and LUE and divided it into lagging URI ($H < -0.1$), synchronous development ($|H| \leq 0.1$), and lagging LUE ($H > 0.1$), where $H = U_1 - U_2$ [52,53].

### 3.4. The Non-Parametric Kernel Density Estimation to Reflecting the Temporal Pattern of CCD

The kernel density estimation is a useful tool for analyzing changes in distributional dynamics, polarization trends, distributional extensibility of the coordinated development of urban and rural areas, and LUE in the YRB, etc. This article utilized a non-parametric kernel density estimation method, which is expressed as follows [21,50]:

$$f(q) = \frac{1}{mh} \sum_{j=1}^{m} K\left(\frac{q_i - q}{h}\right) \tag{8}$$

where $f(q)$ is the kernel density function, $q_i$ is the observation, $q$ is the mean, $h$ is the bandwidth, and $K$ is the kernel function, and this study uses the Gaussian kernel function.

**Table 4.** Classification of the coupled coordination degree (CCD) level.

| CCD Level | Coupling Coordination Stages | Coupled Coordination Features |
|:---:|:---:|:---:|
| $0.3 \leq D < 0.4$ | Moderate disorder | Lagging URI<br>Synchronous development<br>Lagging LUE |
| $0.4 \leq D < 0.5$ | Mild disorder | Lagging URI<br>Synchronous development<br>Lagging LUE |
| $0.5 \leq D < 0.6$ | General coordination | Lagging URI<br>Synchronous development<br>Lagging LUE |
| $0.6 \leq D < 0.7$ | Moderate coordination | Lagging URI<br>Synchronous development<br>Lagging LUE |
| $0.7 \leq D < 0.8$ | Good coordination | Lagging URI<br>Synchronous development<br>Lagging LUE |
| $0.8 \leq D \leq 0.9$ | Good quality coordination | Lagging URI<br>Synchronous development<br>Lagging LUE |

*3.5. The Geographic Detector for Identifying Key Factors*

This study used the geographic model to investigate the factors that affect the coordination degree between URI level and LUE in the YRB. The geographic model has the advantages of a minor sample size limitation and is good at dealing with type volume. Furthermore, the article employed the detector's factor detection and interaction detection to uncover how various drivers and their interactions impact the coupling coordination degree. The specific announcement is as follows (Equation (9)) [1,54]:

$$q = 1 - \frac{1}{n\sigma^2} \sum_{h=1}^{L} n_h \sigma_h^2 \tag{9}$$

where $L$ represents the variable stratification; $n$ and $n_h$ are the number of samples for the whole area and the layer $h$, respectively; and $\sigma^2$ and $\sigma_h^2$ are the sample variances of the entire area and the layer $h$, respectively. In particular, $q$ is the degree of explanation of the detected factor, with a value range between 0 and 1, and $q$ represents the degree of explanation for the detected factor.

## 4. Analysis of Results

*4.1. Evolution of Spatial and Temporal Patterns of Integrated Urban–Rural Development*

Taking the stages of urban–rural relationship adjustment (2003–2006), urban–rural relationship coordination (2007–2011), promotion of urban–rural unity (2012–2017), and the new stage of integrated development (2017-present) as a reference, the ArcGIS 10.2 software was used to determine the URI level in the YRB in 2003, 2007, 2012, 2018, and 2021 (Figure 2) [55]. During the study period, the spatial difference in URI development in the YRB was not apparent, with an average value of 0.379 for upstream prefectures, 0.408 for midstream prefectures, and 0.403 for downstream areas. There were 31 areas with higher than average values, with two regions located in the upstream areas, accounting for 14.29% of the upstream areas; 18 areas in the middle region, accounting for 65.38% of the midstream areas; and 11 downstream regions, accounting for 57.14% of the downstream areas. From the point of view of spatial distribution, from 2003 to 2021, the areas with higher levels of URI development in the YRB were mainly distributed in the following geographical areas: first, in a part of the middle reaches of the YRB consisting of the Baotou-Bayannur-Wuhai-Erdos-Taiyuan-Yuncheng area, which is the main coal resource-rich area,

and second, in the lower reaches of the YRB, in the Dongying-Zibo-Weifang city cluster of the Shandong Peninsula.

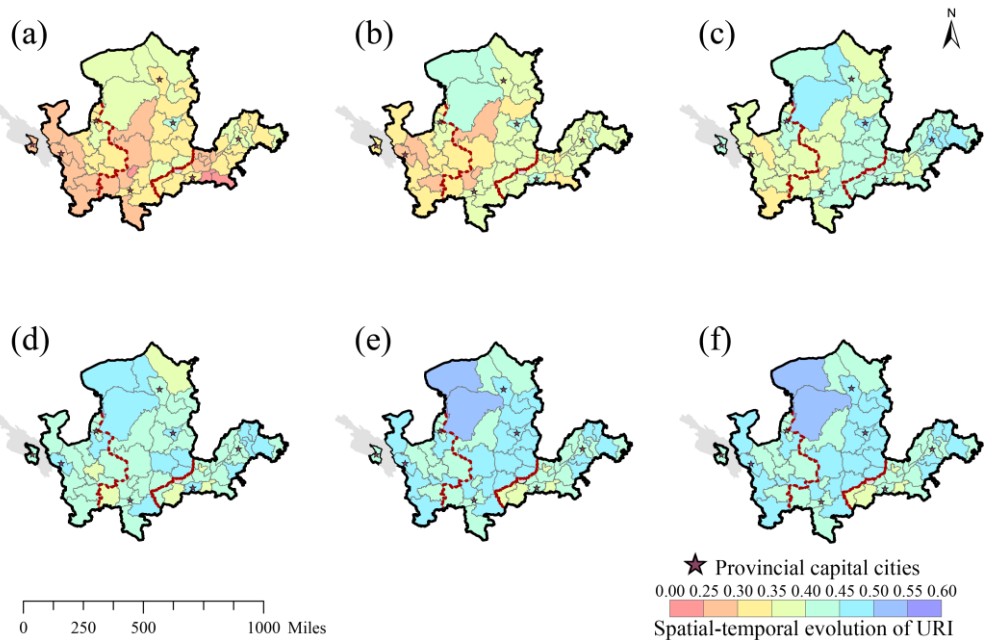

**Figure 2.** Spatial differentiation of the urban–rural integrated (URI) development. (**a**) The URI level in 2003; (**b**) the URI level in 2007; (**c**) the URI level in 2012; (**d**) the URI level in 2018; (**e**) the URI level in 2021; and (**f**) the average URI level.

From the time series evolution (Figure 3), the average value of the URI level in the YRB increased from 0.324 in 2003 to 0.446 in 2021, with an average annual growth rate of 1.91%; the highest annual growth rate was 4.45% in 2010, and the growth rate slowed down significantly in 2016 and 2021. Overall, the standard deviation of the URI level in the YRB decreased from 0.041 in 2003 to 0.032 in 2021, and the regional imbalance in the URI level improved. During the study period, the standard deviation of integrated development in the upstream areas of the YRB was flat, while that in the midstream areas decreased steadily, and in the downstream regions, it initially decreased and then increased. In summary, the mean of the URI levels in the YRB's upper, middle, and lower reaches all improved, and the uneven distribution in URI levels in the midstream areas was alleviated. However, the regional differences in the URI levels in the upstream and downstream areas still needed more attention.

### 4.2. Evolution of Spatial and Temporal Patterns of LUE

Regarding spatial patterns (Figure 4), there were apparent spatial differences in LUE in the YRB. During the study period, the average values of the upstream, middle, and downstream prefectures were 0.451, 0.508, and 0.301, respectively. In total, 29 areas had LUE values higher than the average, with 6 in the upper reaches, accounting for 42.86% of the upstream prefecture-level cities; 17 were in the midstream areas, accounting for 65.38% of the middle reaches; and 6 were in the downstream areas, accounting for 28.57% of the downstream areas. During the study period, the prefecture-level city with the highest LUE in the YRB was Guyuan City in Ningxia, with an LUE of 1.068 in 2021 and an average annual increase of 7.33%. In terms of spatial distribution, during the period of 2003–2021, the areas with a higher LUE value in the YRB were mainly concentrated in the following regions: in part of the midstream areas of the YRB, in the Guyuan-Qingyang-Tongchuan-Longnan-Xianyang-Xi'an area, the Great Guanzhong City Cluster, and Bayannur-Wuhai-Ordos-Hohhot-Yulin-Yanan area, in which are the resource-rich areas; the second is part of the downstream areas of the YRB, the Shandong Peninsula City Cluster, which is dominated by Jining and Jinan.

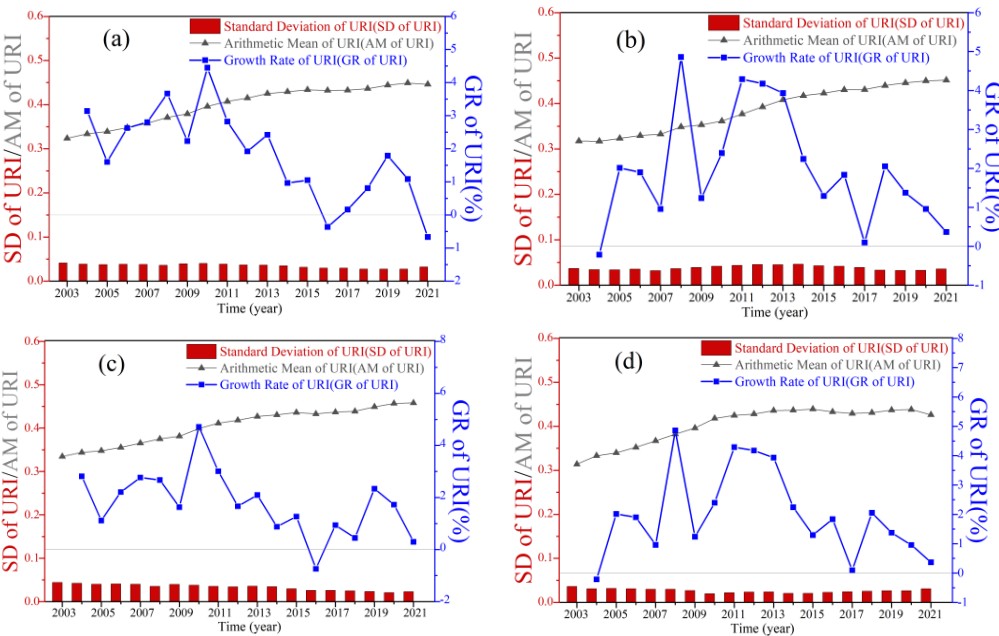

**Figure 3.** Temporal evolution of the URI development from 2003 to 2021. (**a**) The YRB; (**b**) the upstream areas; (**c**) the midstream areas; and (**d**) the downstream areas.

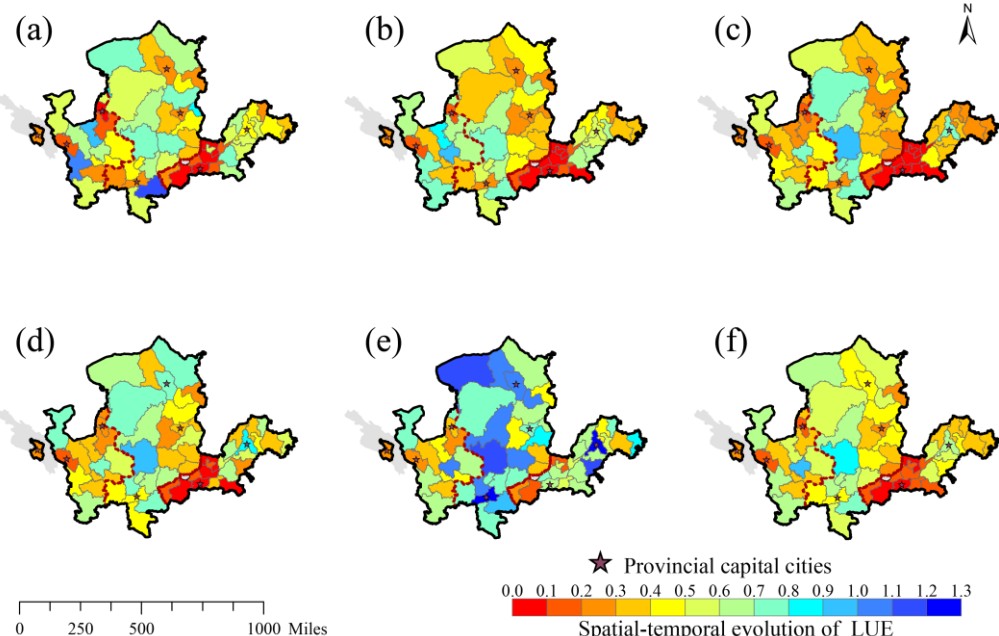

**Figure 4.** Spatial differentiation of the land-use efficiency (LUE). (**a**) The LUE in 2003; (**b**) the LUE in 2007; (**c**) the LUE in 2012; (**d**) the LUE in 2018; (**e**) the LUE in 2021; and (**f**) the average LUE level.

In terms of the time series evolution (Figure 5), the average value of LUE of prefecture-level cities in the YRB increased from 0.456 in 2003 to 0.677 in 2021, with an average annual growth rate of 2.10%, and the highest annual growth rate was 23.41% in 2021. Overall, the standard deviation of LUE in the YRB increased slightly from 0.250 in 2003 to 0.285 in 2021, and the imbalance in LUE among prefecture-level cities had yet to be effectively alleviated. During the study period, the standard deviation of LUE in the upstream, midstream, and downstream areas of the YRB decreased slightly, remained flat, and increased with a fluctuation, respectively. During the study period, the LUE of the upper, middle, and lower reaches of the YRB had improved, and the imbalance in LUE in the upstream areas had been

mitigated. However, the regional differences in LUE in the midstream and downstream areas still needed more attention.

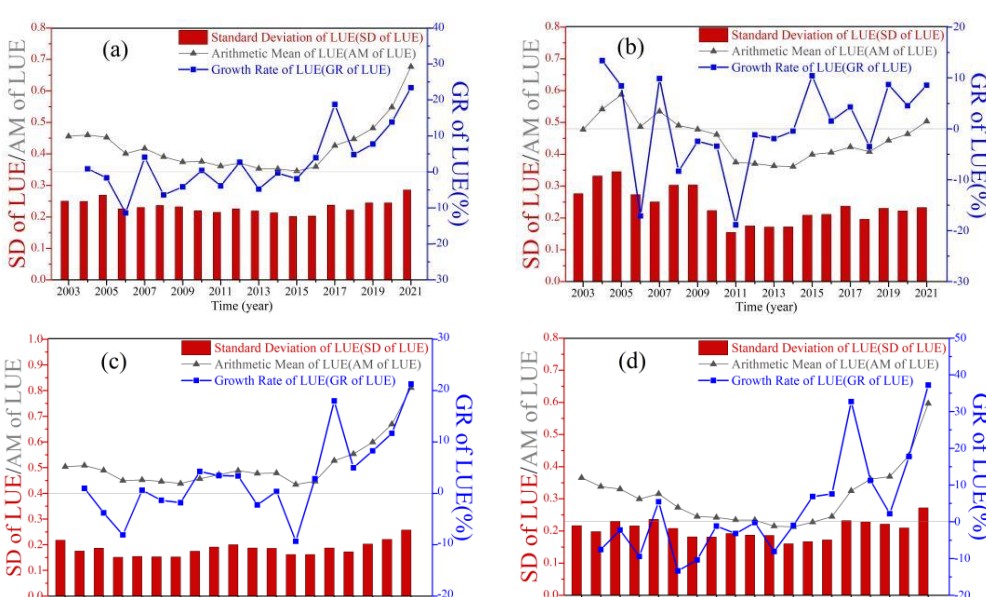

**Figure 5.** Temporal evolution of the LUE from 2003 to 2021. (**a**) The YRB; (**b**) the upstream areas; (**c**) the midstream areas; and (**d**) the downstream areas.

### 4.3. Evolution of Spatial and Temporal Patterns of Coupled Coordination

In terms of a spatial pattern (Figure 6), there were apparent spatial differences in the level of coupled coordination between URI development and LUE; the mean value of upstream, middle-reach, and downstream prefecture-level cities was 0.623, 0.662, and 0.552, respectively. During the study period, the prefecture-level city with the highest level of coupled coordination in the YRB was Wuhai City in Inner Mongolia, with a CCD level of 0.881 in 2021. From the viewpoint of spatial distribution, from 2003 to 2021, the areas with higher coupling coordination degrees were mainly concentrated in the following regions: first, in the midstream areas of the YRB, in the Wuhai-Erdos-Hohhot-Suozhou-Xinzhou-Yulin-Yanan area, which is the primary energy-resource-rich area of the Loess Plateau; and second, in the downstream regions of the YRB, in the Jinan-Dezhou-Jining central urban agglomeration of Shandong Peninsula. Jinan, Hohhot, and Xi'an had better-coordinated development among the provincial capital cities.

From the perspective of the time series evolution (Figure 7), the kernel density curve of coupling coordination degree in the YRB changed significantly. The kernel density curve peak moved to the right and changed from a double peak to a single peak. That is, the overall CCD fluctuated upward, and the bipolar differences gradually narrowed, which is a characteristic of dynamic convergence. At the same time, the main obstacle to improving the CCD in the YRB was the lagging LUE (Figure 8). In the upstream areas, the wave peak of the nuclear density curve moved to the left and then to the right. The peak value increased, decreased, and then increased again, which means that the CCD in the upstream areas showed a fluctuating upward trend. The regional imbalance still needs continuous attention, and the LUE lag dominated the coupling coordination characteristics. The peak of the nuclear density curve in the middle reaches moved to the right, and the right trailing was shortened. The CCD in the midstream areas showed a rising trend, and the inter-regional imbalance was eased, with the LUE mainly lagging behind the coupling coordination features. The kernel density curve in the downstream area changed from a double peak to a single peak, and the height of the wave peak rose gradually, i.e., the polarization phenomenon had been effectively alleviated, the regional disparity had been steadily reduced, and the lagging of the urban–rural integration and development level

dominated the coupling and coordination features. At the same time, the scope of the balanced development area had been expanded.

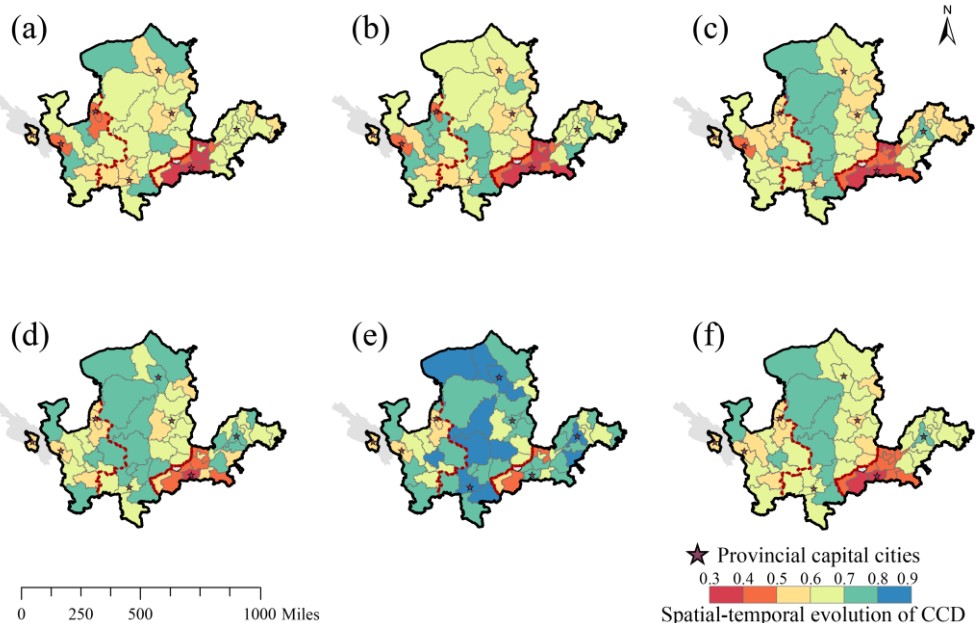

**Figure 6.** Spatial differentiation of coupling coordination degree (CCD) between URI development and LUE. (**a**) The CCD in 2003; (**b**) the CCD in 2007; (**c**) the CCD in 2012; (**d**) the CCD in 2018; (**e**) the CCD in 2021; and (**f**) the average CCD level.

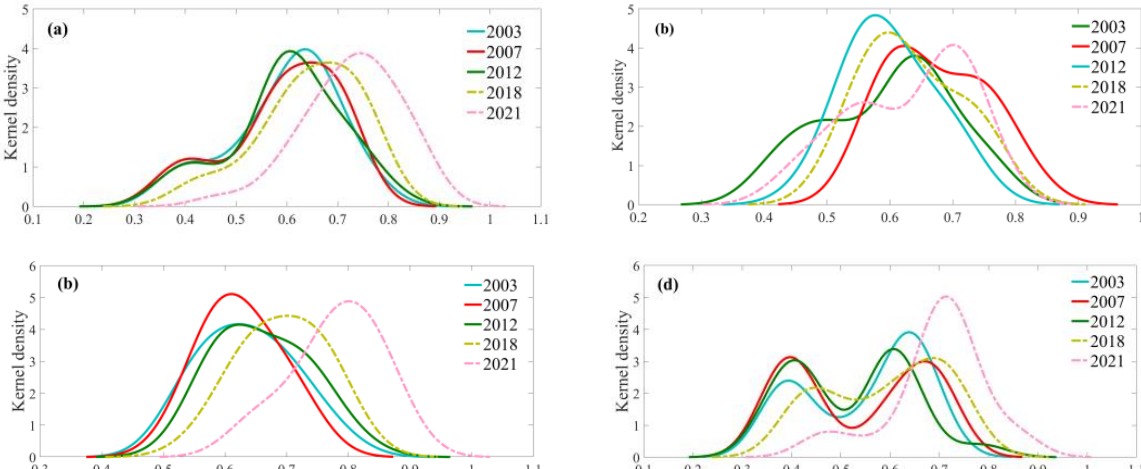

**Figure 7.** Temporal evolution of CCD between URI development and LUE. (**a**) The YRB; (**b**) the upstream areas; (**c**) the midstream areas; and (**d**) the downstream areas.

### 4.4. Analysis of Influencing Factors

A combination of factors affected the degree of coordinated development between the level of URI and LUE in the YRB. Combining the actual situation of the YRB with the research results of several experts and scholars, this study selected ten indicators to investigate from the four aspects of topography, economic level, natural environment, and industrial structure. These indicators were precipitation, elevation, slope, carbon emissions, GDP per capita, urbanization rate, population density, percentage of days with good air quality, per capita arable land area, and the proportion of non-agricultural industries [49,54]. Firstly, multiple linear regression analysis was used to screen the influencing factors, and it was found that six main indicators, with precipitation, altitude, carbon emissions, GDP

per capita, urbanization rate, and population density passed the test at a significance level of 0.01 (Table 5).

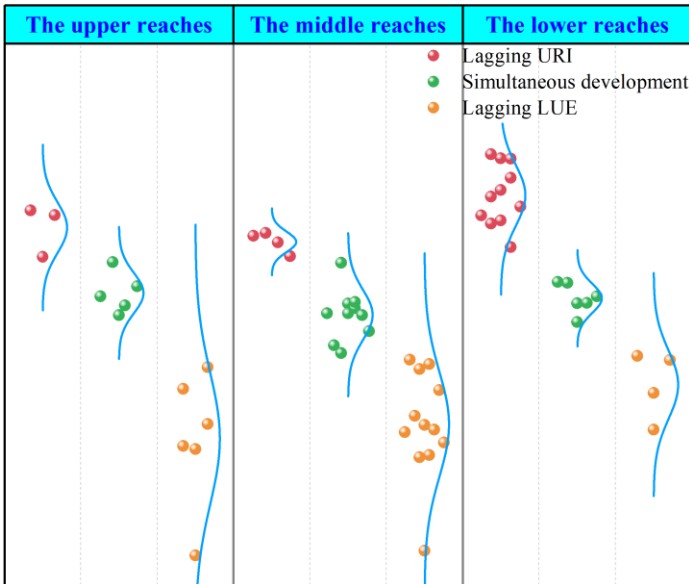

**Figure 8.** Map of the distribution of the characteristics of CCD in the upstream, midstream, and downstream areas of the YRB.

**Table 5.** Statistical table of multiple regression results.

| Variants | Regression Results | Standard Error | Values |
|---|---|---|---|
| Precipitation | 21.222 *** | −7.111 | 1159 |
| High-altitude | 0.023 *** | −0.004 | 1159 |
| Slope | 0.001 | −0.001 | 1159 |
| Carbon emissions | −0.033 *** | −0.007 | 1159 |
| GDP per capita | 0.038 *** | −0.004 | 1159 |
| Urbanization rate | −0.002 *** | 0.000 | 1159 |
| Population density | −0.049 *** | −0.003 | 1159 |
| Percentage of days with good air quality | −0.000 * | 0.000 | 1159 |
| Per capita arable land area | 0.001 | −0.001 | 1159 |
| The proportion of non-agricultural industries | 0.005 | −0.003 | 1159 |
| Constant | 0.791 *** | −0.077 | |

Note: * $p < 0.1$, *** $p < 0.01$.

This study selected the geodetector model to detect the six main driving factors affecting the coupled coordinated development level divergence between URI development and LUE in the YRB in different periods to determine the degree to which each indicator in different periods affects URI development and LUE. The six drivers were classified into five levels using the natural breakpoint method in ArcGIS. The geographic detector detects the factors affecting the spatial variability of the coupled coordinated development level. The contribution and interaction results of each driver are shown in Figures 9 and 10.

According to Figure 10, the $q$ values of the six drivers were 0.131, 0.286, 0.229, 0.181, 0.179, and 0.104 (in this study, a degree of influence $q \geq 0.100$ indicates a highly significant factor). During the study period, the degree of influence of precipitation (X1) on the coordinated development between URI development and LUE in the YRB increased from 0.106 to 0.183, with a mean value of 0.131; the degree of influence of elevation (X2) remained unchanged, with a mean value of 0.286; the degree of impact of carbon emissions (X3) increased from 0.200 to 0.250; that of per capita GDP (X4) increased from 0.143 to 0.180; urbanization rate (X5) increased from 0.068 to 0.294; and population density (X6) decreased from 0.128 to 0.034.

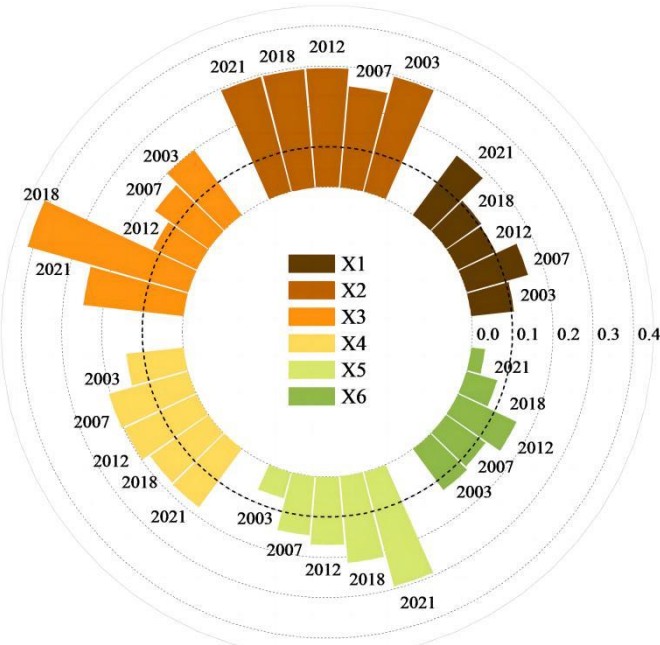

**Figure 9.** Temporal evolution of the degree of influence of CCD between URI and LUE in the YRB. Note: annual precipitation (X1), high-altitude (X2), carbon emissions (X3), per capita GDP (X4), urbanization rate (X5), and population density (X6).

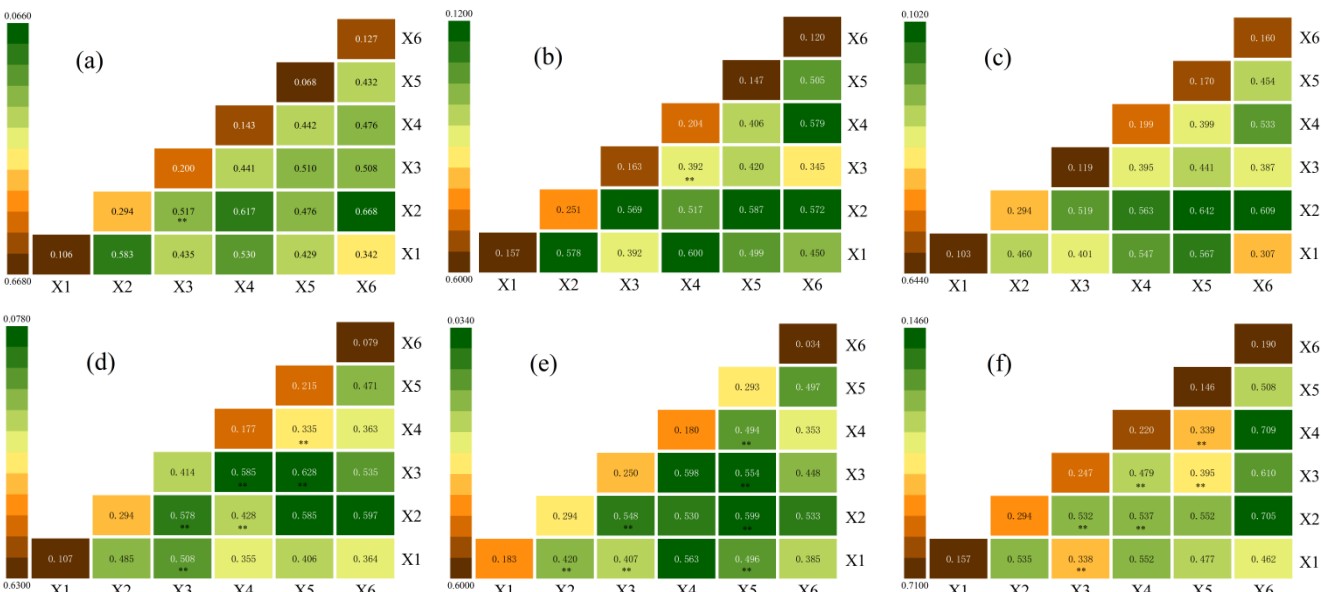

**Figure 10.** Heat map of factors influencing the CCD between URI and LUE in the YRB. The heat map in (**a**) 2003; (**b**) 2007; (**c**) 2012; (**d**) 2018; (**e**) 2021 (**f**) average level. Note: the heat map with ** represents bi-factor enhancement, the remaining are non-linear enhancement.

According to Figure 11, the different drivers have different degrees of influence on the level of coupled development between the level of URI and LUE in the YRB. At the same time, these drivers have a specific interaction relationship. Some had mostly a bi-linear or non-linear enhancement of the interaction during the study period. Based on the results of the interaction analysis of the six drivers and the factor effect strength values, the carbon emission rate (X3), the GDP per capita (X4), and the rate of urbanization (X5) were the three most influential drivers.

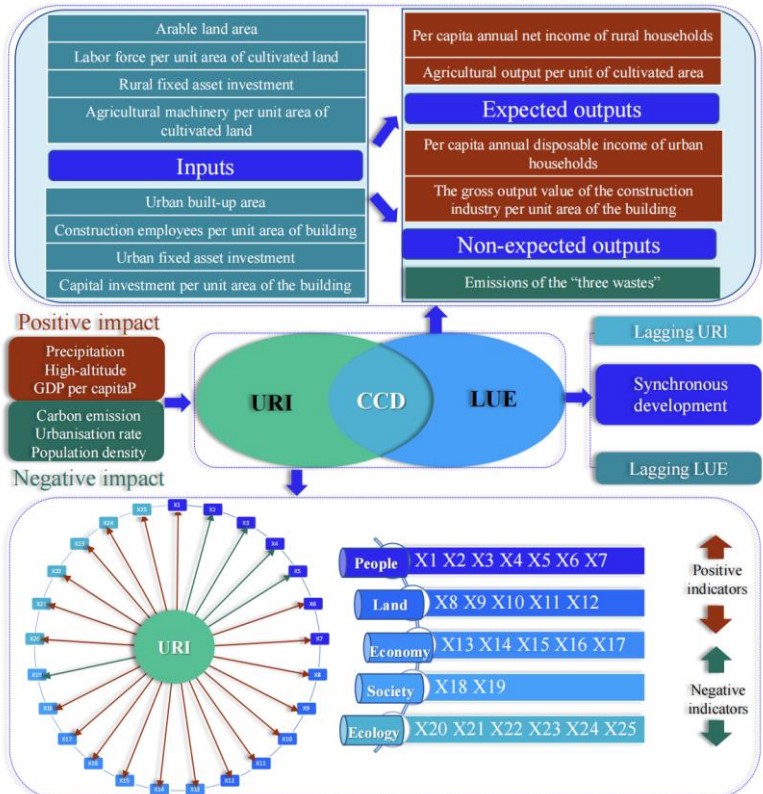

**Figure 11.** Mechanism of the CCD between URI and LUE in the Yellow River. Note: the specific meanings of X1–X25 are shown in Table 2.

## 5. Discussion

The URI development means treating cities and villages as a whole and addressing the imbalances in infrastructure, economic development, and basic public services, and the ecological environment in urban and rural development by promoting the equal exchange of urban and rural factors and reconfiguring the spatial structure of urban and rural areas [7,56]. The land is a critical element of urban and rural development and an essential spatial carrier to promote China's new urbanization construction, achieve comprehensive rural revitalization, and ensure China's URI development [19,57]. Along with China's urban–rural development transformation in the new era, China's urban–rural human–land relationships are undergoing a major restructuring, and the construction of new towns and cities, rural revitalization, and URI development can all be seen as a process of spatial expression of human–land relationships [58]. Focusing on the effective use of land resources, the Chinese government has adopted a variety of means to promote the rationalization of the adjustment of human–land relationships. The first is by promoting the mechanism of "pegging the link between increase and storage" to enhance the economical and intensive utilization of urban land, solve the constraints on urban land-use, accelerate the transformation and upgrading of urban industries, and promote the progress of new types of urbanization [58]. The second is to take the comprehensive improvement of land-use in the whole region as a handhold, improve the protection of arable land and the conservation and intensive use of land, and promote the comprehensive revitalization of the countryside [59]. The third method is to deepen the reform of the land market system, improve the unified urban and rural land-use market, and improve the level of integrated development of urban and rural areas [60]. In 2019, the Chinese government proposed to reshape the urban–rural relationship and solve many problems in land-use in urban–rural development by establishing and improving systems, mechanisms, and policy systems for integrated urban–rural development [13]. This paper studied the CCD and influencing factors of URI development and LUE in the YRB (Figure 11), which tried to make up for

the lack of research on the coupling coordination relationship between URI development and LUE in prefecture-level cities in the YRB, and to deeply explore the internal reasons for the incoordination between URI development and LUE by studying the driving factors.

*5.1. Strengthening Policy Support Is Conducive to Urban and Rural Areas' Comprehensive, Integrated Development and Improving LUE*

Generally, during the study period, the time series evolution of URI level in the YRB showed a fluctuating upward trend [61,62]. The Chinese government has made active policy adjustments for the integrated development of urban and rural areas. In the new era, the Chinese government has changed the urban and rural development strategy from "cities leading rural areas" to "combining urban and rural areas". Strategic policies such as the "New Urbanization Strategy" proposed in 2014 and the "Rural Revitalization Strategy" presented in 2017 can effectively promote the level and quality of URI development [5,16]. During the study period, the regional imbalance in the level of the 61 prefecture-level cities in the YRB has improved. The YRB has seen the most rapid development of urban–rural population integration and urban–rural economic integration, in which the urbanization rate, urban–rural fixed-asset investment ratio, and the level of regional economic performance have improved considerably over the study period. Relatively speaking, prefecture-level cities with a high URI are concentrated in resource-rich areas, such as Shandong Peninsula. On the one hand, since resource-based regions tend to lead in economic development, a higher level of economic development is conducive to the spillover effect of cities on the countryside and to the promotion of integrated urban–rural development [61]. On the other hand, Shandong, as a coastal province with good economic development, has a good level of economic development that can promote the agglomeration and diffusion of resource factors, and the prefecture-level cities in the Shandong Peninsula region have flat terrain, a good agricultural base, and a high level of rural development. At the same time, Shandong Province is also a national-level comprehensive pilot area for the transformation of old and new kinetic energy, which provides a series of favorable conditions for integrated urban–rural development [1]. However, the growth rate of downstream prefecture-level cities was slower than that of the middle and upper reaches, mainly due to the excellent foundation of downstream prefecture-level cities' urban and rural development levels and the small space for progress [61,63]. On the other hand, during the study period, the LUE in the YRB showed an increasing trend with time, and there was spatial differentiation. The continuous improvement of LUE in prefecture-level cities in the YRB was mainly due to the growing attention given by the Chinese government to land resources; it proposed economizing and intensively using existing land resources, improving LUE, and alleviating the contradiction between land supply and demand. From the spatial distribution level perspective, prefecture-level cities with a high LUE are concentrated in resource-based areas, which have been towns seeking breakthroughs in urban transformation and industrial upgrading in recent years, and in medium-sized cities with a high proportion of land redevelopment. At the same time, these cities have better financial support policies for land-use [63,64].

*5.2. The Economic and Intensive Use of Land Resources Is Conducive to the Simultaneous Development of Urban–Rural Integration and Land-Use*

During the study period, the CCD of URI development and LUE in the YRB showed an upward trend over time, the inter-regional imbalance was alleviated, and the regional unevenness in the URI level improved. Lagging LUE was the main obstacle to improving the CCD of URI development and LUE [45,60]. During the study period, the prefecture-level city with the lowest mean value of CCD between URI development and LUE in the YRB was Zhengzhou, Henan Province, located in the midstream areas of the YRB, which also showed a decreasing trend over the study period. This is a result of the depletion of land resources, but the deeper reason is irrational urbanization and untimely policy adjustments [65,66]. During the study period, among the 61 prefecture-level cities in the YRB, the cities with high coordination levels of URI and LUE and relatively fast growth were concentrated in the

resource-rich areas in the middle reaches of the YRB, the urban agglomeration of Shandong Peninsula in the lower reaches, and the provincial capital cities of Jinan, Hohhot, and Xi'an. As these prefecture-level cities are committed to eliminating backward production capacity, developing emerging industries, and optimizing economic structure, they have a relatively high level of URI and abundant financial support funds, which can provide strong support for balanced urban–rural development and land-use [1,48].

*5.3. High-Quality Economic Development, Rational Urbanization Development, and High-End Green Transformation of Industries Increase the CCD between URI Development and LUE*

The CCD of URI development and LUE in the YRB were affected by many factors. First, the increase in carbon emissions significantly negated urban and rural development. The critical elements in the increase in carbon emissions are rapid urbanization development, and a substantial increase in population and industrial production activities that need to be environmentally friendly. The YRB is also known as the "Energy Basin", and the long-term development of traditional high-pollution, high-water-consumption, and high-energy-consumption industries has led to a sharp rise in carbon emissions in the YRB, and studies have proved that excessive carbon emissions affect the efficiency of urban–rural integration and development, as well as ecological protection and high-quality development in the YRB [67]. In response to this, the Chinese government issued a policy in 2022 to achieve a low-carbon transition in energy consumption through green industrial development. Therefore, the Chinese government has proposed a "new urbanization strategy" and "high-end green transformation of industries" [68]. In addition, as the world's most significant carbon dioxide ($CO_2$) emitter, China needs to reduce carbon emissions to achieve "carbon neutrality" by 2050 [69,70]. Second, the increase in per capita GDP (X4) had a significant positive impact. Regions with higher per capita GDP have more fiscal revenue, which is conducive to improving the balance between urban and rural development, protecting land carbon storage, and improving LUE [13,71]. Third, the urbanization rate (X5) had a significant negative impact. As some prefecture-level cities in the YRB are located in ecologically sensitive areas such as the Loess Plateau, the existing studies have shown that unreasonable urbanization development will lead to low sustainability of land-use and unbalanced urban–rural development. At the same time, the urbanization of some cities is promoted by encroaching on wetlands and lakes, or even destroying pristine mountain ranges. Therefore, attention must be paid to sustainable new urbanization and rural revitalization in urbanization development [72–74]. The development of urbanization in the sensitive areas of the Loess Plateau can learn from the advanced experience of the large urban engineering, for example, the "Mountain Excavation and City Construction (MECC)" in China [75].

In conclusion, existing studies have largely advanced our comprehension of the spatio–temporal heterogeneities and changing dynamics of various aspects related to URI development, land-use efficiency, and institutional systems in the YRB. However, our findings may have the following limitations: First, this study analyzed the coupled and coordinated relationship and influencing factors between urban–rural integration and LUE, and concluded that there is an interactive correlation between urban–rural integration and LUE. Unfortunately, there are few theoretical discussions and empirical tests on the intrinsic mechanism of these two aspects in the existing literature, and this article only made a preliminary exploration of this topic. On the one hand, URI development can promote the intensive and economical use of land resources through optimizing and upgrading the industrial structure of urban and rural areas, enhance the economic benefits of land, and then effectively improve the efficiency of land-use. At the same time, URI development can induce the free flow of urban and rural factors, and optimize the allocation of land resources, so as to achieve the purpose of enhancing the efficiency of land-use. In addition, urban–rural integration can help strengthen the construction of transport and information networks and water and electricity infrastructure, enhance regional competitiveness, and ultimately effectively improve LUE. On the other hand, improving LUE can provide land

factor security for urban–rural integration development by increasing the economic density of land, and then increasing the economic carrying capacity per unit of land. At the same time, LUE can help optimize the industrial spatial layout to achieve rational allocation of land and clustering of economic space, and thus promote the integrated development of urban and rural areas. In addition, the improvement in LUE can reasonably adjust the relationship between urban and rural populations and land, promote the two-way flow of urban and rural populations, enhance the employment level of urban and rural residents, and ultimately effectively enhance the level of URI development. However, this study failed to further deepen the research because the exploration of the inner mechanism of URI development and LUE requires more complex logical deduction on the basis of a solid theory, and needs to be tested and verified with the help of empirical data. This aspect is also the direction of future research on URI development and LUE in the YRB. Second, only ten influencing factors were selected for analysis in this study, which may not sufficiently represent the actual situation. Since the existing research focuses on the unilateral research on urban–rural integration or LUE, and there is less research on the influencing factors on the relationship between urban–rural integration and LUE, this study mainly referred to the indicators in the existing literature that have an impact on both relationships when selecting indicators. At the same time, we also considered the actual situation of the YRB to select the indicators. For example, foreign investment has an impact on LUE, but it is less related to URI development, so this indicator was not selected [76]. In addition, the research on URI development and LUE covers a wide range, and the influencing factors affecting their relationship are complex and diverse, but the selection of influencing factors needs to take into account the availability and accuracy of information and data. Generally speaking, authoritative data come from the data published by the statistical department, but the current statistical data have differences in the statistical calibre and scope, and the lack of temporal and comprehensive information data limits the selection of influencing factors. In future research, further field research and questionnaire surveys can be carried out to obtain first-hand research information through in-depth collection of basic data at the meso- and micro-levels, so as to better carry out the research on the factors influencing urban–rural integration development and LUE. Third, a significant positive effect of precipitation growth was found in the regression analysis, but the article does not provide extensive explanations on this matter. The main reason for this is that the exploitation and consumption rate of surface water in the YRB has far exceeded the carrying capacity of the Yellow River's water resources, and at the same time, most of the YRB is in an arid and semi-arid region, and the ecological environment relies on atmospheric precipitation to a high degree. Worse still, the annual precipitation and the number of annual rainfall days in the YRB have been on a declining trend in recent years, leading to an increasingly serious shortage of water resources in the YRB [77,78]. The shortage of water resources has become a major contention in the tensions between man and land, and has seriously constrained the high-quality development of urban and rural areas in the YRB [79]. However, the results of the regression analyses are only used as a preliminary screening of influencing factors, and the influence of precipitation will not be discussed in depth here.

## 6. Conclusions

Based on existing studies, this paper discussed the coupling and coordination relationship between URI development and LUE in the YRB. This study adopted the linear weighting method, super-efficiency SBM model based on non-expected output, CCD model, non-parametric kernel density estimation method, and geographical detector to explore the dynamic evolution characteristics and influencing factors of URI development and LUE in the YRB from 2003 to 2021.

The results showed that: (1) the spatial distribution of urban–rural integration in the YRB maintained a balanced level, showing the spatial distribution characteristics of "blurred difference boundaries, relatively high in the middle and lower reaches". During

the study period, the regional imbalance of urban–rural integration in the YRB as a whole still needs continuous attention, and the regional imbalance of urban–rural integration in the middle reaches had been significantly alleviated. The overall LUE showed a spatial distribution characteristic of "high at both ends and low in the middle". The regional imbalance of LUE in the upstream region was somewhat alleviated during the study period. However, the regional inequality of LUE in the middle and downstream areas still needs continuous attention. (2) There were apparent spatial differences between the CCD of URI and LUE, similar to the distribution of LUE, showing the spatial distribution characteristics of "high at two ends and low in the middle". A high level of coupling coordination was mainly observed in the middle and lower reaches of the YRB. During the study period, the CCD of the YRB showed a fluctuating upward trend, and the regional imbalance was alleviated; in particular, the provincial inequalities in the middle and lower reaches were effectively alleviated. The main characteristic of CCD in the YRB were the lag in LUE. (3) According to the analysis of influencing factors, it can be concluded that carbon emissions, per capita GDP, and urbanization rate significantly impact the CCD of URI development and LUE. The increased carbon emissions and improved urbanization rate had a significant negative impact on URI development and LUE. The growth of per capita GDP had a significant positive impact on URI development and LUE, and a small number of the driving factors had a bilinear enhancement effect, although most of the driving factors had a nonlinear enhancement effect. In future research, on the one hand, we can evaluate districts and counties, carry out field research and questionnaire surveys, and study the URI development and LUE in a small region by collecting basic data at the meso- and micro-levels. This will help the local government to implement the optimal allocation strategy of land-use that is closer to the reality, to achieve more efficient land resource use, and promote the integrated development of urban and rural areas. On the other hand, with the deepening of the research on URI development and LUE, more and more related research will appear; on this basis, the internal mechanism of URI development and LUE can be explored in depth and tested and verified with the help of empirical data from the existing research.

**Author Contributions:** Conceptualization, C.S. and Q.W.; methodology, C.S.; validation, L.S. and X.W.; formal analysis, L.S. and X.W.; resources, X.W. and J.D.; data curation, C.S.; writing—original draft preparation, C.S.; writing—review and editing, C.S., L.S., X.W. and J.D. All authors have read and agreed to the published version of the manuscript.

**Funding:** This research was funded by National Natural Science Foundation of China, grant numbers: "42271221" "42061037".

**Data Availability Statement:** The data presented in this study are available on request from the corresponding author.

**Conflicts of Interest:** The authors declare no conflict of interest.

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
