# Peer review of "Exploring the Coupling Coordination and Key Factors between Urban–Rural Integrated Development and Land-Use Efficiency in the Yellow River Basin"

_land, doi:10.3390/land12081583_

Round 1
Reviewer 1 Report
1. In the Introduction, the literature review is mostly about the current research progress of Chinese scholars. It is recommended to conduct a comprehensive review to summarize the international progress.
2. What is the contribution of this article? This should be clearly stated to show the innovation of the paper.
3. Line 155, “the YBR includes 61 geospatial units……”, it should be “YRB……”. Line 243, “in the YBR”, it should be “in the YRB”. And in the caption of Figure 3, the same problem exists. There have been several spelling errors, please check it.
4. In Table 1, there are blank spaces between table and heading.
5. Figure 7 is not clear enough. It is recommended that the three-dimensional curve be modified to a two-dimensional curve to better show the variation in the kernel density profile.
6. In the section 4.4, the coefficient of precipitation on Coupled Coordination Degree is 21.222. How to explain that precipitation has such a large influential role?
7. In Table 4, please check the variable names, “Percentage of days with air Quality higher than Grade 2”.
8. What is the purpose of Figure 9? There is no corresponding explanation for the spatial distribution of the variable’ effect. So, the authors should consider whether it is necessary to present Figure 9.
Extensive editing of English language required. There are some particularly long sentences in the paper and some grammatical errors.
Author Response
Dear Revierwer:
We sincerely thank the reviewer for thoroughly examining our manuscript and providing very helpful comments to guide our revision. We have tried our best to revise the manuscript according to your kind and construction comments and suggestions. Please find the following detailed responses to your comments and suggestions.We sincerely hope that this revised manuscript has addressed all your comments and suggestions. We have provided a point-by-point response to the reviewers' comments below in red color. Meanwhile, in the latest MS Word, there are also traces of modification and specific modification contents. Our response to the questions are shown below.

Reviewer 2 Report
The manuscript describes the importance of exploring the complex dynamic relationship between integrated urban-rural development and land-use efficiency, and elaborates on the idea of how to establish a sound and unified urban-rural land-use market for urban and rural construction and rationally promote integrated urban-rural development.
1. Lines 98-123: There are no obvious grammatical problems with this passage. However, it contains some longer sentences and more complex structures. Further editing and revision may be needed to improve readability and clarity.
2. Lines 163-164: Figure 1 does not have latitude and longitude coordinates. Is the map of China appropriate? There is no nine-dash line. Scale and latitude/longitude coordinates should also be provided.
3. The content of the manuscript image is not clear; it is recommended that it be adjusted appropriately, e.g., font size.
4. 165-172 lines: It is recommended to change to a table. Write down the name of the data, source and treatment, etc.
5. 189 and 239 lines: table names 2 and 3 are bold or not, please standardise.
6. 229 lines: formula (7), is the C-value calculated correctly?
7. 329 and 346 lines: figure names 2 and 4 are incorrect, please align.
8. Lines 428-430: Please further explain the key role of land in urban and rural development. How does it contribute to China's new urbanisation, comprehensive rural revitalisation and ensure URI development?
9. Lines 450-453: What are the main areas of improvement in URI levels in the 61 prefecture-level cities in the Poyang Lake Basin during the study period? In particular, what are the specific characteristics of prefecture-level cities in resource-based regions and Shandong Peninsula region?
10. Lines 490-493: What are some specific examples or evidence of the impacts of increased carbon emissions on urban and rural development and land use efficiency mentioned in this passage?
11. Lines 511-525: In addition to the ten influences that have been selected, what other influences might have an impact on integrated urban-rural development and land-use efficiency? Why have they not been analysed quantitatively? Can some relevant justifications or explanations be provided?
12. Lines 511-525: Based on the suggestions in the text, how should future research enhance the comprehensive study of integrated urban-rural development and land use in the Yellow River Basin? What aspects need to be focused on in order to explore the linkages and mechanisms of coordinated urban-rural integrated development and land use efficiency?
13. At the end of the conclusion, could you specify how future research should further strengthen the integrated study of urban-rural integrated development and land use efficiency?
14. The references are very inappropriate and do not correspond to the text paragraphs and need to be double-checked and revised. For example, 28?
Author Response

(The authors gave the same response as above.)

Reviewer 3 Report
Figure 1 modify legend because Basin boundary is Middle Basin Boundary
Can you explain how you boundary this area? By altitude for example?
Table 1 Indicators are not clean, you have to write better the formula for example x6, x13 is wrong GDP per capita (GDP Yuan/person), YUAN in PPP (Purchasing power parity) do you consider the role of inflation?
Equation 1 are you sure it is correct? Is it always zij??
Equation 2 is not entropy, is the share so The entropy of the indicator j is calculated as shown as Equation (2) and (3) began The entropy of the indicator j is calculated as shown as Equation (3) where pij is calculate in equation 3
Author Response

(The authors gave the same response as above.)

Round 2
Reviewer 2 Report
All problems have been modified and agreed to accept.
Reviewer 3 Report
it's ok for me thanka